# Ternary Blends from Biological Poly(3-hydroxybutyrate-*co*-3-hydroxyhexanoate), Poly(propylene carbonate) and Poly(vinyl acetate) with Balanced Properties

**DOI:** 10.3390/polym15214281

**Published:** 2023-10-31

**Authors:** Yujie Jin, Changyu Han, Yi Li, Hongda Cheng, Dongdong Li, Huan Wang

**Affiliations:** 1School of Materials Science and Engineering, Jilin Jianzhu University, Changchun 130118, China; 2Key Laboratory of Polymer Ecomaterials, Changchun Institute of Applied Chemistry, Chinese Academy of Sciences, Changchun 130022, China

**Keywords:** biodegradable polymers, poly(3-hydroxybutyrate-*co*-3-hydroxyhexanoate), poly(propylene carbonate), poly(vinyl acetate), mechanical properties, polymer blends

## Abstract

Poly(3-hydroxybutyrate-*co*-3-hydroxyhexanoate) (PHBH) has gained significant attention because of its biodegradability and sustainability. However, its expanded application in some fields is limited by the brittleness and low melt viscoelasticity. In this work, poly(vinyl acetate) (PVAc) was introduced into PHBH/poly(propylene carbonate) (PPC) blends via melt compounding with the aim of obtaining a good balance of properties. Dynamic mechanical analysis results suggested that PPC and PHBH were immiscible. PVAc was miscible with both a PHBH matrix and PPC phase, while it showed better miscibility with PHBH than with PPC. Therefore, PVAc was selectively localized in a PHBH matrix, reducing interfacial tension and refining dispersed phase morphology. The crystallization rate of PHBH slowed down, and the degree of crystallinity decreased with the introduction of PPC and PVAc. Moreover, the PVAc phase significantly improved the melt viscoelasticity of ternary blends. The most interesting result was that the remarkable enhancement of toughness for PHBH/PPC blends was obtained by adding PVAc without sacrificing the strength markedly. Compared with the PHBH/PPC blend, the elongation at the break and yield strength of the PHBH/PPC/10PVAc blend increased by 1145% and 7.9%, respectively. The combination of high melt viscoelasticity, toughness and strength is important for the promotion of the practical application of biological PHBH.

## 1. Introduction

Interest focusing on biodegradable biopolymers has exponentially increased over the past few decades on account of the growing plastic accumulation and depletion of fossil resources. Poly(3-hydroxyalkanoate) (PHA) is one of aliphatic bio-polyesters produced from the microorganism fermentation of sugar or plant-based oils, which has attracted much industrial and scientific attention due to its biodegradability, biocompatibility and renewability of raw materials [1,2]. So far, over 150 kinds of PHAs composed of different co-monomers have been synthesized, but only a few of them have the potential to be produced commercially, such as polyhydroxybutyrate (PHB), poly(hydroxybutyrate-*co*-hydroxyhexanoate) (PHBH) and poly(hydroxybutyrate-*co*-hydroxyvalerate) (PHBV) [3].

PHBH is a thermoplastic copolymer containing two different monomers, namely the short-chain 3-hydroxybutyrate (3HB) units and the medium-long-chain 3-hydroxyhexanoate (3HHx) units [4]. The 3HB/3HHx copolymerization ratio can affect the physical properties of PHBH with a range from amorphous elastomer to semicrystalline plastic. Therefore, PHBH is expected to play a role in multiple fields, such as food packaging, disposable plastics and agricultural mulch films [5,6]. However, PHBH with low 3HHx content is brittle, which originated from its high crystallinity, which limits its practical applications. Therefore, it is necessary to improve the flexibility of PHBH while maintaining its elastic modulus and mechanical strength. Polymer blending is an effective approach with a low production cost to overcome polymer drawbacks and combine characteristics of blend components. Blending PHBH with other biodegradable polymers can improve the performance and is in line with the sustainable development strategies and eco-friendliness concepts. Many biodegradable PHBH-based blends have been reported. For example, Lim et al. [7] prepared immiscible PHBH/polycaprolactone (PCL) blends and found that the PCL phase improved the yield strength and modulus of PHBH/PCL blends. Blending poly(butylene adipate-*co*-terephthalate) (PBAT) with PHBH inhibited the crystallization of PHBH and improved the processability and mechanical properties [8]. Yu et al. [9] found that the miscibility of PHBH/poly(ethylene oxide) (PEO) blends was dependent on the blend composition, and the mechanical properties of PHBH were improved by the introduction of 5–17.5 wt% PEO. PHBH was miscible with poly(3-hydroxybutyrate-*co*-4-hydroxybutyrate) (P34HB), and the PHBH/P34HB blends displayed higher thermal stability, flexibility and tensile strength compared with neat PHBH [10].

Poly(propylene carbonate) (PPC), as an aliphatic polycarbonate, is made by alternating copolymerization of carbon dioxide (CO_2_) with propylene oxide, whose synthesis process mitigates CO_2_ accumulation in the environment [11]. The outstanding properties of PPC, including good processability, biodegradability and ductility, can render it of much industrial interest for applications in packaging, foams and agricultural mulch films. More recently, the utilization of rare earth catalysts allowed for the large-scale synthesis of PPC, which made it easy to utilize PPC at low cost [12,13]. PPC had been blended with a number of biodegradable polymers, for instance, poly(lactic acid) (PLA) [14,15], PHBV [16], PHB [17] and poly(butylene succinate) (PBS) [18], to enhance their mechanical properties. Zhang et al. [19] investigated the PHBH/PPC solution blend films with different weight ratios. It was observed that PHBH matrix and PPC phase were immiscible in the melted state. As PPC content was increased from 0 to 50 wt%, the elastic modulus decreased from 92.9 to 69.6 MPa, and the elongation at the break increased from 16% to 20%.

For thermodynamically immiscible polymer blends, lower interfacial interactions and poorer phase morphology limit the combination of the advantages of blend components. Therefore, compatibilization is required for an immiscible PHBH/PPC blend to improve the phase morphology and give the blends desired properties. The most common approaches to enhance the compatibility of immiscible polymer blends are the addition of block or graft copolymers, or “reactive blending” [20]. The main limitations of block or graft copolymers are the limited availability and high cost. Reactive blending is possible when the polymer components being blended can be modified by mutually reactive functional groups and remain structurally stable under processing conditions. Therefore, another method of introducing the third polymer component into immiscible blend systems to improve their compatibility has attracted attention. Moussaif et al. [21] reported that the introduction of polymethylmethacrylate (PMMA) into immiscible polycarbonate (PC)/polyvinylidenefluoride (PVDF) polymer blends decreased interfacial tension and improved tensile properties. Zhang et al. [22] prepared ternary PLA/PHBV/PBS blends with a good balance of toughness and stiffness and found that PHBV and PLA were partially miscible, while PBS was immiscible with PLA or PHBV.

Poly(vinyl acetate) (PVAc) with amorphous characteristics was reported to be miscible with PHB [23], PCL [24] and PLA [25]. Gao et al. [26] revealed that in the PPC/PLA/PVAc ternary blends, PVAc was selectively localized at the interface between the PLA dispersed phase and PPC matrix, which served as compatibilizers to increase interfacial adhesion and improve the mechanical properties of the blends. Therefore, in the present study, in view of the complementary property between PHBH and PPC, coupled with the miscibility between PVAc, PHBH and PPC, PVAc was selected as the third polymer phase to be introduced into the PHBH/PPC blends. Furthermore, the effect of PVAc concentration on the dynamic mechanical property, phase morphologies, melting and crystallization, tensile properties and rheological properties was investigated in detail. It was expected that PVAc phase could improve the compatibility of the PHBH/PPC blends and refine the phase structure, which resulted in an improvement in the mechanical and rheological properties of PHBH/PPC/PVAc blends.

## 2. Materials and Methods

### 2.1. Materials

PHBH was provided by Jiangsu Lansu Biomaterial Co., Ltd. (Yancheng, China). It had a weight-average molecular weight (*M*_w_) and polydispersity (PDI) of 754 kg mol^−1^ and 4.89, respectively. PHBH copolymer had 6 mol% 3HH unit based on ^1^H NMR. PPC was provided by Changchun Institute of Applied Chemistry. It had an *M*_w_ of 130 kg mol^−1^ and PDI of 2.6. PVAc with *M*_w_ of 315 kg mol^−1^ and PDI of 1.9 was bought from Nuoda New Materials Company (Yantai, China). Chemical structures of the three polymer components are provided in Figure 1.

### 2.2. Sample Preparation 

To remove moisture, before processing, PHBH, PPC and PVAc were dried at 80 °C under vacuum for 10 h. Melt blending of samples with different PVAc content was carried out with an internal mixer (XSS300, Shanghai Kechuang Rubber Plastic Mechanical Equipment Co., Ltd., Shanghai, China). Blending parameters included temperature of 160 °C, screw speed of 60 rpm and residence time of 7 min. The blended samples were hot-pressed at 170 °C and under a fixed pressure of 10 MPa for 3 min, then cold-pressed at room temperature and 10 MPa for 3 min to obtain 1-mm thick samples for tests. The weight composition of PHBH/PPC = 70 wt%/30 wt% was fixed in binary and ternary blends. For the PHBH/PPC/PVAc blends, PVAc contents were 5, 10 and 20 wt% of the total ternary blends. It was worth noting that neat PHBH, neat PPC, PHBH/PPC binary, PPC/PVAc binary and PHBH/PVAc binary blends undergoing the same thermal processing were also prepared as control samples. The PVAc weight content was set as 30 and 50 wt% in the PPC/PVAc binary blends and as 10, 20 and 30 wt% in the PHBH/PVAc binary blends. 

### 2.3. Characterizations

#### 2.3.1. Dynamic Mechanical Analysis (DMA)

Dynamic mechanical property tests were conducted on a DMA850 from a TA Instruments (New Castle, DE, USA) at oscillating amplitude of 5 μm and frequency of 1 Hz. The tests were carried out at temperatures ranging from −30 to 100 °C with a heating rate of 3 °C min^−1^. The rectangle sample dimension was 20 × 10 × 1.0 mm^3^.

#### 2.3.2. Rheological Measurements

A rheometer (AR2000EX, TA Instrument, USA) with a parallel-plate geometry (diameter = 25 mm) was performed to investigated the melt viscoelastic properties of neat PHBH and its blends. The dynamic frequency sweep was carried out at 155 °C and swept from 100 to 0.05 rad s^−1^ under dry nitrogen. The fixed strain of 1.25% ensured that the response was within the linear viscoelastic range.

#### 2.3.3. Scanning Electronic Microscopy (SEM)

The phase morphologies of neat PHBH and all blends were observed using a field emission scanning electron microscopy (XL30 ESEM FEG, FEI Co., Hillsboro, OR, USA) with an accelerating voltage of 10 kV. All sample films were cooled in liquid nitrogen for 20 min, then were cryogenically fractured. The smooth fractured surface of neat PHBH was taken for direct SEM observation, and all blends were etched by immersing them in acetone solution for 120 min, ensuring the dissolution of PPC phase but the preservation of PHBH matrix. Prior to SEM observation, the surfaces of all samples were sputter-coated with gold. The average size of dispersed phase (*D*) of the blends was calculated using Nano Measurer 1.2 software and analyzing 50–100 particles of SEM images for each sample. 

#### 2.3.4. Differential Scanning Calorimetry (DSC)

Melting and crystallization behaviors of neat PHBH and all its blends were probed by TA Instruments DSC (Q20, USA) under N_2_ atmosphere. Around 5−8 mg of samples were heated from 40 to 190 °C at a heating rate of 100 °C min^−1^ and left at 190 °C for 3 min to erase the thermal history. The first cooling scan was monitored from 190 to −30 °C at a rate of 5 °C min^−1^ for determining the crystallization temperature (*T_c_*) and crystallization enthalpy (Δ*H_c_*). After that, the second heating scan was performed between −30 and 190 °C at a heating rate of 10 °C min^−1^, from which the glass transition temperature (*T_g_*), cold crystallization temperature (*T_cc_*), cold crystallization enthalpy (Δ*H_cc_*), melting temperature (*T_m_*) and melting enthalpy (Δ*H_m_*) could be obtained. The degree of crystallinity (*X_c_*) of PHBH in all blends was calculated with the following formula:(1)$$X_{c}=\frac{\Delta H_{m}-\Delta H_{cc}}{\Delta H_{m}^{0}\alpha }\times 100\%,$$
where $\Delta H_{m}^{0}$ represents the fusion enthalpy of 100% crystalline PHBH (146 J g^−1^) [27], and *α* represents the weight fraction of PHBH in the blends. It was worth noting that all enthalpy values including Δ*H_c_*, Δ*H_cc_* and Δ*H_m_*, were corrected based on the weight content of PHBH in the blends. 

For the isothermal crystallization analysis, all samples were first heated at a rate of 100 °C min^−1^ from 40 to 190 °C and held for 3 min, then quickly cooled at a rate of 45 °C min^−1^ to the desired crystallization temperature. Finally, the samples were further isothermally crystallized at the crystallization temperature for the sufficient time.

#### 2.3.5. Tensile Tests 

The tensile properties of neat PHBH and all its blends were measured at room temperature by using an Instron-1121 tensile tester (Canton, MA, USA). The gauge length of samples was 20 mm, and crosshead speed was set to 10 mm min^−1^. At least 5 bars for each specimen were measured under the same conditions to obtain an average value. 

## 3. Results and Discussion

### 3.1. Dynamic Mechanical Analysis

In order to investigate the mutual miscibility of the three polymer components, DMA tests were performed. This is because the phase structure and morphology of immiscible blends depend on the interfacial interactions and miscibility between polymer components. The damping factor (tan *δ*) curves of neat PHBH, PPC, PVAc, PHBH/PPC binary blends and PHBH/PPC/PVAc ternary blends are shown in Figure 1a. In order to study the miscibility between components more clearly, tan *δ* curves of PPC/PVAc binary and PHBH/PVAc binary blends are given in Figure 1b and 1c, respectively.

The tan *δ* peak is directly related to the transition in molecular mobility, which represents the glass transitions of polymer. It is well known that the glass transition temperature (*T_g_*) of the polymer blend system is an important criteria to assess the miscibility of polymer blends. If the blend has only one *T_g_*, between the two neat components, it means that the two components of the blend are completely miscible. If the blend shows two composition-independent *T_g_*s, which are close to the *T_g_*s of neat components, it suggests that the blend is completely immiscible. Two composition-dependent *T_g_*s, which are between those of neat components, indicates a partially miscible polymer blend [28]. Table 1 shows *T_g_* values of neat components and all blends. As shown in Figure 1a and Table 1, sharp tan *δ* peaks were observed at about 13.9 °C for neat PHBH, 33.0 °C for neat PPC and 47.9 °C for neat PVAc, corresponding to their glass transitions. For the PHBH/PPC binary blend, two *T_g_*s, one at around 12.5 °C, close to that of neat PHBH, and the other at around 34.7 °C, close to that of neat PPC, were observed, suggesting that PHBH and PPC were immiscible. Moreover, such a phenomenon was also in accordance with the literature report [19]. For the PHBH/PPC/PVAc blends, the damping peaks of PPC were located at about 35 °C and did not show significant changes with increasing PVAc content. The increase in the *T_g_* of the PHBH matrix was observed for PHBH/PPC/5PVAc and PHBH/PPC/10PVAc blends, and the *T_g_* of PHBH in the PHBH/PPC/10PVAc blend was around 6 °C higher than that of PHBH in the PHBH/PPC binary blend. The *T_g_* of PHBH in the PHBH/PPC/20PVAc blend was hardly detected due to the fact that the *T_g_* of PHBH was close to that of the PPC phase. As can be seen from Figure 1b, PPC/PVAc binary blends showed only one tan *δ* peak, which was between those of neat PPC and PVAc and increased with PVAc content. A variation of the *T_g_* of PHBH/PVAc binary blends was similar to that of PPC/PVAc blends, as shown in Figure 1c. These results demonstrated that PVAc was miscible with both PHBH and PPC and displayed better miscibility with PHBH compared with PPC. Therefore, it can be inferred that PVAc was probably selectively localized in the PHBH matrix, increasing the *T_g_* of PHBH in the PHBH/PPC/PVAc blends. 

Figure 2 shows the temperature dependence of storage modulus (*E*′) of neat components, the PHBH/PPC blends and PHBH/PPC/PVAc blends. Of the three neat glassy components at −10 °C, neat PHBH exhibited the highest *E*′ of about 5.2 GPa, and neat PPC had the smallest *E*′ of about 2.9 GPa. Due to the glass transition, *E*′ of PHBH, PPC and PVAc decreased sharply at about 10, 30 and 50 °C, respectively. In case of PHBH/PPC binary blend, *E*′ was between that of neat PHBH and that of neat PPC. For ternary blends, PVAc phase had little influence on *E*′ of the PHBH/PPC/PVAc blends at temperature below *T_g_* of PVAc (about 50 °C). However, *E*′ of the PHBH/PPC/PVAc blends decreased significantly with the increase in PVAc content above the *T_g_* of PVAc. Such a result was possibly owing to the dilution effect of PVAc on the PHBH matrix.

### 3.2. Rheological Properties

Rheology is often used to investigate the melt behavior of non-Newtonian fluids, such as polymers. The viscosity of a Newtonian fluid changes with temperature and does not change with the strain rate. However, a non-Newtonian liquid displays a change in viscosity with the strain rate. Therefore, the investigation of rheology is particularly important for determining the fluid mechanics of polymer [29]. The rheological properties of polymer blends are essential for understanding the relationships between the processibility and structure-property of polymer blends. Figure 3 shows the curves of the storage modulus (*G*′), loss modulus (*G*″), complex viscosity (|*η**|) and damping factor (tan *δ*) of neat PHBH, PPC, PVAc, the PHBH/PPC blends and the PHBH/PPC/PVAc blends at 155 °C. 

According to *G*′ in Figure 3a, neat PHBH had much lower melt elasticity than those of neat PVAc and PPC. Neat PVAc showed higher melt elasticity at low and intermediate frequency regions and lower melt elasticity at high frequency regions compared to that of neat PPC. The binary and ternary blends displayed a lower *G*′ compared to neat PPC and a higher *G*′ compared to neat PHBH. The increase in the PVAc content gradually increased the melt elasticity of ternary blends at low frequency regions due to the fact that PVAc had higher melt elasticity compared with the neat PHBH. Similar to *G*′, the gradual introduction of the PVAc component into the PHBH/PPC blend also raised the *G*″ of the blends (Figure 3b). As shown in Figure 3c, neat PHBH showed a Newtonian plateau at intermediate frequency regions with the smallest viscosity of the three neat components. In the low-frequency region, the decreasing complex viscosity with decreasing frequency might be due to the slight degradation of PHBH during testing at 155 °C. Neat PVAc showed significant shear-thinning behaviour, suggesting that its complex viscosity was a function of frequency. Clearly, with the introduction of PVAc, the Newtonian plateau of the PHBH/PPC/PVAc blends at low frequency regions became progressively smaller, and the complex viscosity increased significantly. The damping factor (tan *δ*) is the ratio of dissipated energy (*G*″) to stored energy (*G*′). The tan *δ* data in Figure 3d demonstrated that PHBH had the largest damping factor among the three neat components and displayed a frequency dependence of tan *δ*, suggesting its viscous liquid characteristic. The tan *δ* values of PVAc were smaller than those of PHBH and PPC and were not very dependent on frequency, suggesting its elastic liquid characteristic. The tan *δ* curves of binary and ternary blends were between those of PHBH and PPC and decreased with an increase in the PVAc content at a low frequency due to the lower tan *δ* values of PVAc compared to those of PHBH. 

As stated earlier, the dynamic viscoelastic properties of the blends implied that the presence of PVAc affected the rheological properties of PHBH/PPC blends. The influence of PVAc content on the rheological properties was due to its localization in the continuous phase of PHBH, which improved the interfacial interactions between the PHBH matrix and PPC dispersed phase, thereby promoting the dispersion of PPC domains in the PHBH matrix. By considering that the melt viscosity and elasticity of PVAc was higher than that of PHBH, adding PVAc into the PHBH/PPC blend resulted in an increase in the entanglement density in the continuous phase and at the interface between the PPC dispersion phase and the PHBH/PVAc continuous phase, as PVAc was mixed with PHBH macromolecular chains at their molecular-scale [30]. The increase in the degree of entanglement density was directly connected with the PVAc content. The enhancement of the entanglement density of the PHBH matrix and between the matrix and dispersion phase resulted in the increased viscosity and elasticity of the melt. This was due to that the increased physical interactions through chain entanglement restricted macromolecular flow and slip under shear deformations. Additionally, the entanglement density in the ternary blends could strongly retard the macromolecular chain relaxations, which might lead to an increase in the melt strength of the PHBH matrix [31]. The improved melt strength was very meaningful for PHBH processing due to one of its drawbacks of a low melt strength. 

### 3.3. Phase Morphology

The phase morphology of blends is a crucial key affecting the macroscopic properties of blend systems. Therefore, the investigation of the size of dispersed phases and morphology types of the PHBH/PPC/PVAc ternary blends in this work was important and helpful for determining the relation between microstructure and resulting performance. Figure 4 gives the SEM micrographs of neat PHBH and all blends. 

As shown in Figure 4(a1,a2), neat PHBH showed featureless and smooth cryo-fractured surfaces. For the PHBH/PPC and PHBH/PPC/PVAc blends (Figure 4(b1–e2)), the dispersed PPC phase etched away by acetone left homogeneous and spherical voids, implying that the PHBH matrix and PPC phase were immiscible, which was in line with the observation of DMA. It was worth noting that the acetone solution was a solvent for neat PVAc. From the previous discussion, PVAc was miscible with PHBH at the molecular level and was selectively localized in the PHBH matrix. Therefore, the small amount of PVAc dispersed in the PHBH matrix could not be solubilized in the acetone solution. It was of great interest that a much finer and more uniform phase structure was observed with the addition of the PVAc phase. Figure 5 clearly demonstrates the dispersed phase size obtained using an analysis of the Nano Measurer 1.2 software. As can be seen from Figure 5, the average diameter (*D*) of the dispersed phase decreased from 1.68 μm for the PHBH/PPC binary blend to a half value for the PHBH/PPC/20PVAc ternary blend (0.84 μm).

The phase morphologies of immiscible polymer blends was dependent on interfacial tension between the phases, blend compositions, processing parameters and viscosity ratio of the components [32,33]. The final morphology of immiscible blends was controlled by the competition between the droplet coalescence and break-up during melt processing [34]. It is well known that for the polymer blends with “sea-island” phase structure, phase coarsening frequently occurs with the coalescence of droplets, which reduces the interfacial area and decreases the free energy of blend systems [35]. The addition of compatibilizers to immiscible polymer blends can control phase morphology and enhance the miscibility of the blends by reducing interfacial tension [36]. Therefore, in the present work, it can be inferred that the PVAc phase in the PHBH/PPC/PVAc blends was fully miscible with the PHBH matrix and PPC phase and was selectively localized in the PHBH matrix, acting as compatibilizers to reduce the interfacial tensions between the PPC phase and PHBH matrix and refine the phase structure. In addition, PVAc as a third phase was selectively located in the PHBH matrix, forming a continuous phase together with the PHBH, which also led to a smaller size of the dispersed phase with the increasing PVAc content. Studies on the influence of the viscosity ratio (the ratio of dispersed phase viscosity to matrix phase viscosity) on the morphology of immiscible polymer blends indicated that the larger the viscosity ratio, the coarser the phase structure, while a viscosity ratio of approximately equal to 1 led to much finer morphology [37,38]. In the present work, as shown in Figure 3c, the viscosity of PPC was significantly greater than that of PHBH, indicating that the PHBH/PPC binary blend had a high viscosity ratio. For the PHBH/PPC/PVAc ternary blends, PVAc with a higher viscosity was selectively dispersed in the PHBH matrix, which could increase the matrix viscosity and reduce the viscosity ratio, resulting in a refined morphology. Similar results were observed in partially miscible PHBH/PLA blends with a reactive epoxy compatibilizer (REC) to enhance the compatibility between PHBH and PLA phases [39].

### 3.4. Thermal and Crystallization Behaviors

A differential scanning calorimetry (DSC) test was performed to study the thermal behaviors of neat components and their blends, as recorded in Figure 6. The thermal parameters are summarized in Table 2. Figure 6a clearly indicated that neat PHBH exhibited a significant crystallization exothermic peak (*T_c_*) at about 50 °C. Neat PPC and PVAc had no crystallization peak, suggesting their amorphous characteristics. For the PHBH/PPC binary blends, it was observed that the *T_c_* of PHBH shifted to a low temperature, suggesting that the crystallization of PHBH was suppressed by adding the PPC phase. Similarly, for the immiscible PHBH/PBAT and PHBH/PBS blends, the presence of PBAT and PBS inhibited the crystallization of PHBH [8,40]. In contrast, PCL was found to promote the crystallization of PHBH in immiscible PHBH/PCL blends [40]. However, for the PHBH/PPC/PVAc blends, no crystallization peaks were observed, suggesting that the presence of PVAc inhibited the crystallization of PHBH. 

As shown in Figure 6b, neat PHBH, PPC and PVAc showed the *T_g_*s at 2.8, 33.3 and 44.7 °C, respectively. The *T_g_* of PPC was not apparent for all blends, probably because the glass transition peaks overlapped with the onset of cold crystallization exothermic peaks. The *T_g_* of PHBH in the blends was increased with increasing PVAc content, which was consistent with DMA observations, suggesting that PVAc was miscible with the PHBH matrix.

From Figure 6b, neat PHBH had no cold crystallization (*T_cc_*), indicating its relatively complete crystallization during the cooling process. Furthermore, the *T_cc_*s of the PHBH/PPC/PVAc blends increased significantly with increasing PVAc content. For instance, the *T_cc_* of the PHBH/PPC blend was 54.4 °C, and it increased to 80.4 °C for the PHBH/PPC/20PVAc blend. The obvious increase in the *T_cc_* of PHBH in the ternary blends once again illustrated that the introduction of PVAc into the PHBH/PPC blend inhibited the crystallization of PHBH. First, PVAc, which was located in the PHBH matrix, would restrict the stacking ability and mobility of PHBH molecular chains due to its high viscosity. Second, the presence of amorphous PVAc increased the entanglement density between PHBH molecular chains, which also inhibited the chains’ mobility [30].

Neat PHBH and all blends showed typical double melting peaks during the second heating process. The melting peaks on the low-temperature side and high-temperature side are labeled as *T*_*m*1_ and *T*_*m*2_, which correspond to the melting endothermic peaks of primary and recrystallized PHBH crystals, respectively [41]. The *T*_*m*1_ of PHBH in the PHBH/PPC blend was lower than that of neat PHBH, which was probably owing to the fact that the PPC phase affected the thickness of the lamellae of the primary crystals of PHBH. The *T*_*m*1_s of PHBH in the PHBH/PPC/PVAc ternary blends were found to increase gradually with increasing PVAc content. This might be due to the fact that the incorporation of PVAc raised the *T_cc_* of PHBH in the ternary blends, leading to the formation of more stable and perfect primary crystals of PHBH. The *T*_*m*2_s of the blends did not change significantly compared to that of neat PHBH. 

From Table 2, it was observed that the degree of crystallinity (*X_c_*) of neat PHBH was 38.3%, and the *X_c_* of the binary blend decreased to 16.6%, which was due to the incorporation of PPC inhibiting the crystallization of PHBH. For the PHBH/PPC/PVAc blends, the *X_c_* continued to decrease with the introduction of PVAc. This was due to a decrease in the number of PHBH segments and chains growing toward the crystal growth front caused by the dilution of PVAc. In addition, the complete miscibility of PHBH and PVAc led to a migration of heterogeneities from the PHBH to the PVAc phase. Consequently, the number of heterogeneous primary nuclei of PHBH was decreased. 

### 3.5. Isothermal Melt Crystallization Behavior and Kinetics

In this section, the influence of PVAc content on the isothermal melt crystallization kinetics of the PHBH/PPC blend was further investigated with DSC. DSC scans of isothermal crystallization of neat PHBH and all blends at 80 and 85 °C are presented in Figure 7. The relative crystallinity (*X_t_*) at crystallization time (*t*) was determined with the following equation [42]:(2)$$X_{t}=\frac{X_{c}(t)}{X_{c}(\infty )}=\int_{0}^{t}\frac{dH(t)}{dt}dt/\int_{0}^{\infty }\frac{dH(t)}{dt}dt,$$
where *X_c_* (*t*) is defined as the crystallization enthalpy at time *t*, and *X_c_* (∞) is defined as the enthalpy when crystallization is finished. The d*H*(*t*)/d*t* is the rate of heat flow during isothermal crystallization at time *t*. The development of *X_t_* versus crystallization time *t* is showed in Figure 8. Obviously, all plots presented the shape of “S”, and the crystallization time of the PHBH/PPC/PVAc blends was prolonged with an increase in the PVAc content. Furthermore, the Avrami equation was used to analyze isothermal crystallization kinetics of PHBH and its binary and ternary blends as follows [43]:(3)$$X_{t}=1-\exp(-kt^{n}),$$
where *n* (*n* = *n*_1_ + *n*_2_) is the Avrami exponent depending on both nucleation type (*n*_1_) and growth dimension (*n*_2_), while *n*_1_ equals 1 for homogeneous nucleation and *n*_1_ equals 0 for heterogeneous nucleation. The variable *k* is the crystallization rate constant, which is associated with both nucleation and growth. Figure 9 presents the Avrami plots of neat PHBH and all blends, and *n* and *k* values obtained from the linear portion of Avrami plots are listed in Table 3. An *R*^2^ value greater than 0.99 indicated that the Avrami equation could describe the isothermal crystallization kinetics of the binary and ternary blend system well. In general, neat polymers have an *n* value of 4 with sporadic nucleation mode, while polymers containing heterogeneous nucleating agents show an *n* value of 3 with instantaneous nucleation [44]. In practice, impurities in the neat polymers result in the heterogeneous nucleation with a three-dimensional spherulite growth at *n* values of 3. From Table 3, regardless of the crystallization temperature and PVAc content, neat PHBH and its blends displayed similar values of *n* in a range of 2.4 to 2.8, indicating heterogeneous nucleation with two-dimensional to spherulitic crystal growth (*n*_1_ = 0, an d *n*_2_ = 2 or 3) [45]. Moreover, the crystallization temperature and blending with PPC and PVAc did not change the crystallization mechanisms of PHBH.

Crystallization half time (*t*_1/2_), defined as the time to achieve 50% *X_t_*, could be used to evaluate the crystallization rate and was calculated with the following equation:(4)t12=(ln2k)1n

The *t*_1/2_ values of neat PHBH and its blends are summarized in Table 3. As can be seen in Table 3, when the crystallization temperature was increased from 80 to 85 °C, the crystallization time of the samples was increased. For example, the *t*_1/2_ of PHBH/PPC/5PVAc at 80 °C was 20.9 min, which was increased to 35.3 min at 85 °C. This was due to the fact that an increase in the crystallization temperature led to a decrease in supercooling, which reduced the driving force and resulted in difficulties in crystallization nucleation [46]. At crystallization temperature of 80 or 85 °C, the *t*_1/2_ of the blends was greater than that of neat PHBH and increased with the increasing PVAc content. These results indicated that the incorporation of PVAc reduced the isothermal crystallization rate of the PHBH matrix, which could be due to the dilution influence of the amorphous PVAc melt on the PHBH crystal region. Another possible reason for the decrease in the crystallization rate of PHBH in the PHBH/PPC/PVAc blends was that the high viscosity of PVAc as well as the increased entanglement density inhibited the diffusion and stacking of PHBH molecular chains [22].

### 3.6. Tensile Mechanical Properties

The stress−strain curves of neat PHBH, the PHBH/PPC blends and the PHBH/PPC/PVAc blends are presented in Figure 10, and the mechanical properties of all samples are summarized in Table 4. Based on Figure 10a, neat PHBH was a rigid polymer and exhibited a brittle fracture fashion with much lower elongation at the break of 4.0%. It could be seen that the PHBH/PPC binary blend also failed in brittle mode, while it showed improved elongation at the break compared to neat PHBH. It could be expected that the modulus and yield strength of the binary blend were smaller than those of neat PHBH. This could be mainly due to the fact that the phase-separated morphology of the PHBH/PPC blend led to the interfacial debonding.

For the PHBH/PPC/PVAc ternary blends, the elongation at the break was increased with increasing PVAc content. The ternary blends with 10 and 20 wt% PVAc underwent distinct yielding, suggesting the occurrence of brittle-to-ductile fracture transition. The ternary blend with 20 wt% PVAc especially showed a considerable and stable cold drawing after yielding, and the elongation at the break increased to 636%, which was more than 150 times higher than that of neat PHBH. It is well known that strength and toughness are important requirements for most structural materials. It is a pity that strength and toughness are usually mutually exclusive. Typically, the toughening of polymers would be accompanied by a sharp decrease in the strength of binary blends. The blends of PHBH also showed such trends [8,19,47]. For example, the elastic modulus and tensile strength of PHBH/PPC (70 wt%/30 wt%) decreased by 11.2% and 0.05%, respectively, and the elongation at the break increased by 23.9% compared to neat PHBH [19]. Katsumata et al. [47] observed that the introduction of a small amount of PCL could increase the toughness of the P(3HB-*co*-7 mol% 3HH) cast film, where the Young’s modulus and the maximum stress descended greatly. However, for the blend of PHBH/PPC/5PVAc in this work, the presence of the PVAc phase increased the strength of PHBH from 31.6 to 34.3 MPa, and the elongation at the break was enhanced from 4 to 19%. With the addition of 10 wt% PVAc in the PHBH/PPC blend, the yield strength of the blend decreased slightly from 31.6 to 30.0 MPa, and a more significant increase in the elongation at the break, up to 84.7%, was observed.

The significantly greater elongation at the break of the PHBH/PPC/PVAc blends compared to neat PHBH was derived from the presence of the PVAc phase in the PHBH matrix, which confined the crystallization of PHBH and significantly reduced the degree of crystallinity due to the entanglement density and the dilution effect of PVAc, as shown in Figure 11. As a result, more amorphous PHBH regions were deformed and oriented when being stretched. On the other hand, the dilution effect of the PVAc phase weakened the intermolecular interactions of PHBH in the PHBH/PPC/PVAc blends, leading to an easier flow of PHBH molecular chains. The PVAc phase selectively localized in the PHBH matrix acted as a compatibilizer, reducing the interfacial tensions and refining the phase morphology, leading to an increase in yield strength.

Figure 12 shows the comprehensive properties of neat PHBH, the PHBH/PPC blends and the PHBH/PPC/PVAc blends, including the yield strength, Young’s modulus, elongation at the break, melt viscosity and storage modulus at an angular frequency of 0.05 rad s^−1^. As can be seen from Figure 12, the incorporation of PVAc into the PHBH/PPC blends significantly improved the flexibility and melt viscoelasticity of ternary blends. Compared to neat PHBH, the melt viscosity, melt elasticity and elongation at the break of PHBH/PPC/10PVAc blend were increased by 126%, 4636% and 2000%, respectively, while the yield strength and Young’s modulus decreased by only 8% and 5%, respectively. Therefore, the significant increase in flexibility and melt viscoelasticity with no deterioration in strength compared to neat PHBH were achieved by adding small amounts of amorphous PVAc into the PHBH/PPC binary blend, which was positive for expanding the applications of PHBH blends. 

## 4. Conclusions

PHBH/PPC/PVAc ternary blends were prepared via a melt blending method with the aim of obtaining multiple good performances. A DMA analysis revealed that PHBH and PPC were immiscible. PVAc was miscible with both the PHBH matrix and PPC dispersed phase and displayed better miscibility with PHBH compared with PPC. Rheological property studies demonstrated that the elasticity and viscosity of blend melts increased significantly with the increasing PVAc content due to the high viscosity and elasticity of the PVAc melt and the increased entanglement density. The SEM results of all blends exhibited phase-separated morphology, and dispersed phase diameter gradually decreased with the increasing PVAc content. The PVAc phase was selectively localized in the PHBH matrix. Based on the DSC results, the PVAc phase suppressed the cold crystallization and reduced the degree of crystallinity of the PHBH matrix. The isothermal crystallization rate of the PHBH/PPC/PVAc blends was significantly decreased with adding PVAc, while the crystallization mechanism did not change. It was of great interest to observe that the introduction of a certain amount of PVAc significantly improved the toughness of the ternary blends and kept the strength without deterioration. This work provided a simple method to tailor the strength, ductility, viscosity and elasticity of the melt and the crystallization rate of the PHBH blends, which represented promise for the realization of specific properties, thus further expanding the application of PHBH.

## Figures and Tables

**Scheme 1 polymers-15-04281-sch001:**
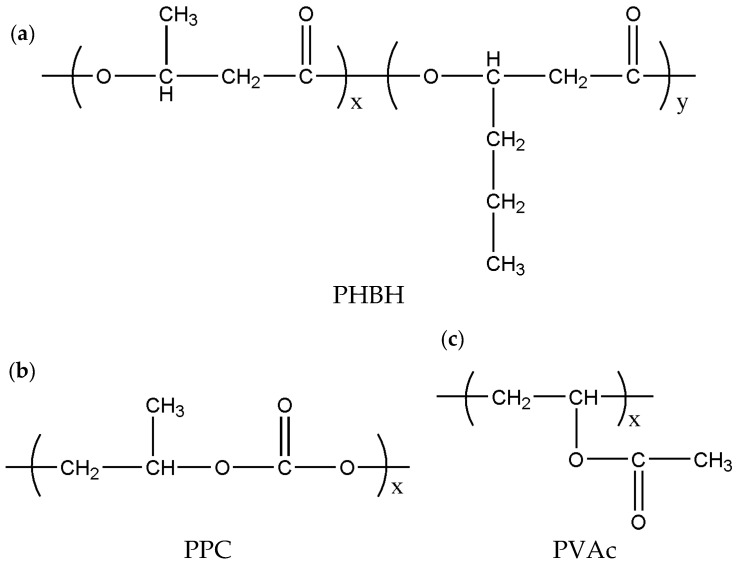
Chemical structures of (**a**) PHBH, (**b**) PPC and (**c**) PVAc.

**Figure 1 polymers-15-04281-f001:**
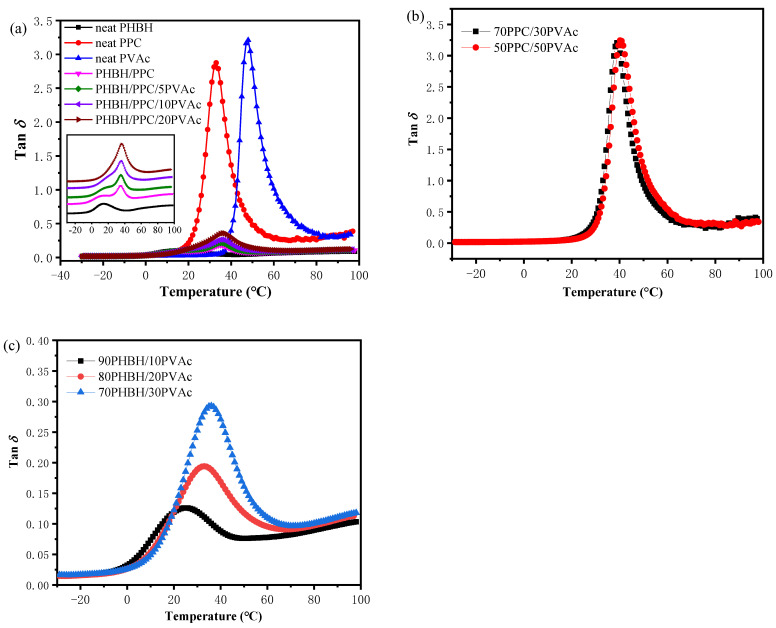
Damping factor (tan *δ*) curves of (**a**) neat PHBH, PPC, PVAc, PHBH/PPC binary and PHBH/PPC/PVAc ternary; (**b**) PPC/PVAc binary and (**c**) PHBH/PVAc binary blends.

**Figure 2 polymers-15-04281-f002:**
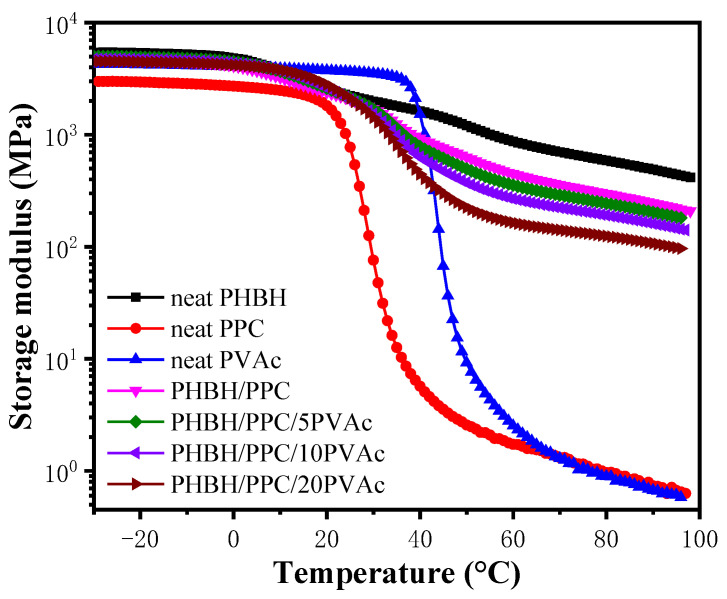
Storage modulus (*E*′) curves of neat PHBH, PPC, PVAc, PHBH/PPC blends and PHBH/PPC/PVAc blends.

**Figure 3 polymers-15-04281-f003:**
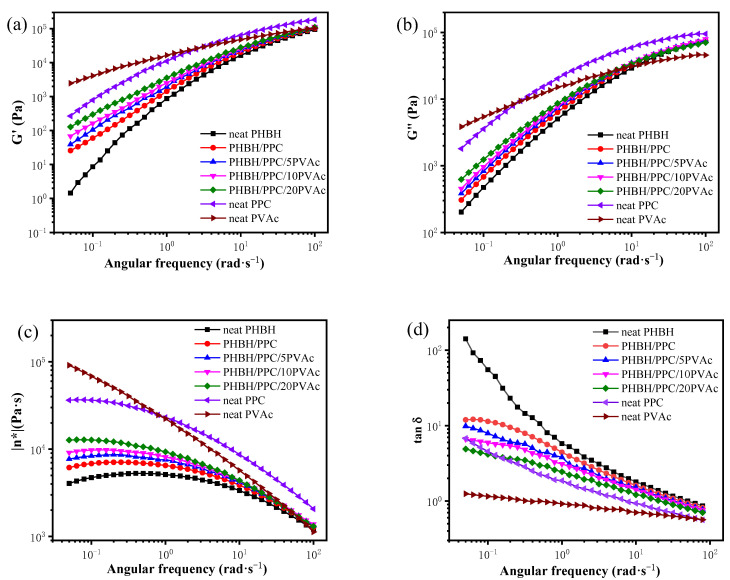
Dynamic viscoelastic properties of neat PHBH, PPC, PVAc, PHBH/PPC blends and PHBH/PPC/PVAc blends at 155 °C: (**a**) storage modulus (*G*′), (**b**) loss modulus (*G*′′), (**c**) complex viscosity (|*η**|) and (**d**) loss tangent (tan *δ*).

**Figure 4 polymers-15-04281-f004:**
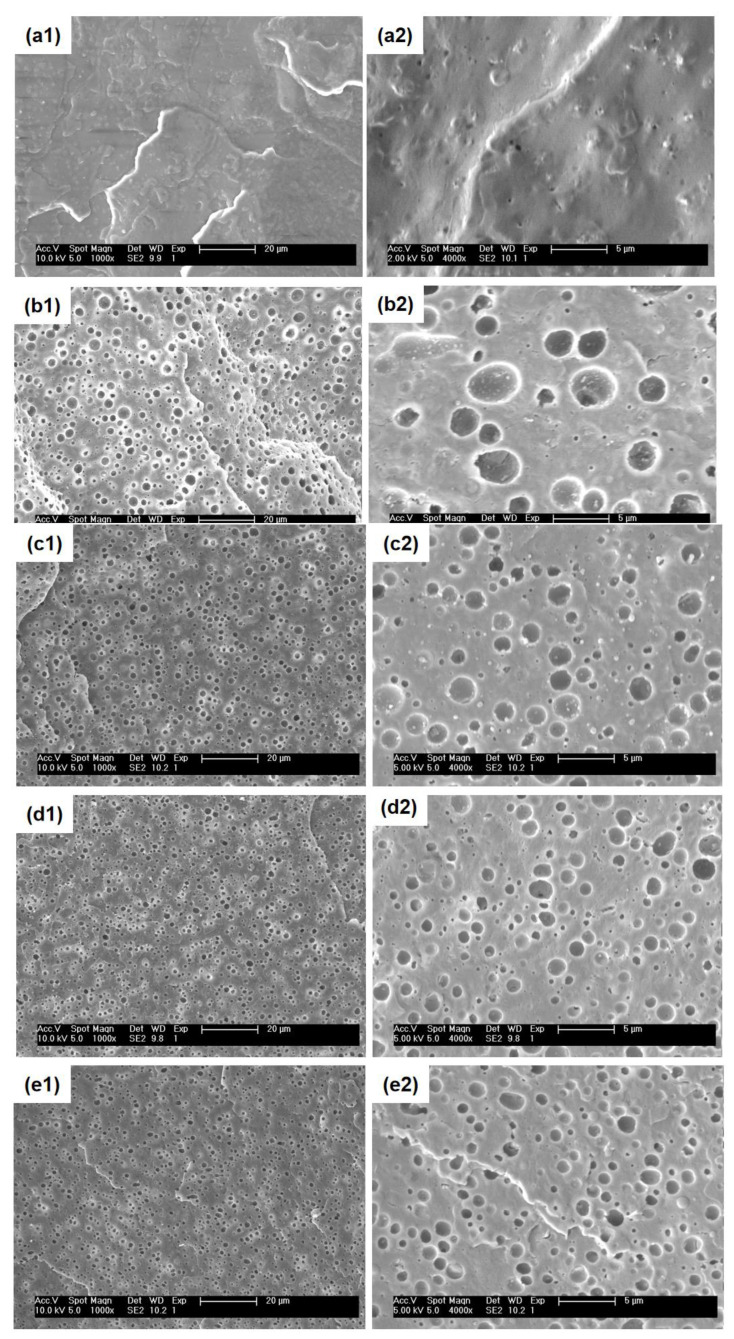
SEM images of surfaces for: (**a1**,**a2**) neat PHBH, (**b1**,**b2**) PHBH/PPC, (**c1**,**c2**) PHBH/PPC/5PVAc, (**d1**,**d2**) PHBH/PPC/10PVAc and (**e1**,**e2**) PHBH/PPC/20PVAc at different magnifications.

**Figure 5 polymers-15-04281-f005:**
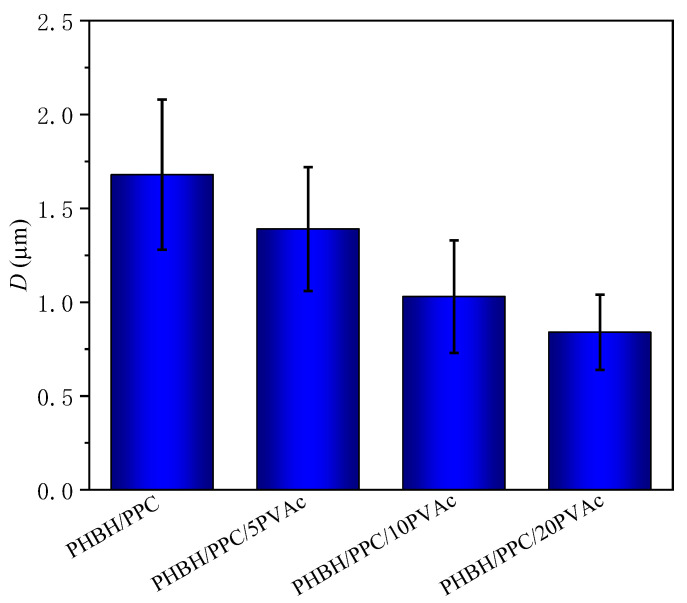
The average size of dispersed phase (*D*) of PHBH/PPC and PHBH/PPC/PVAc blends.

**Figure 6 polymers-15-04281-f006:**
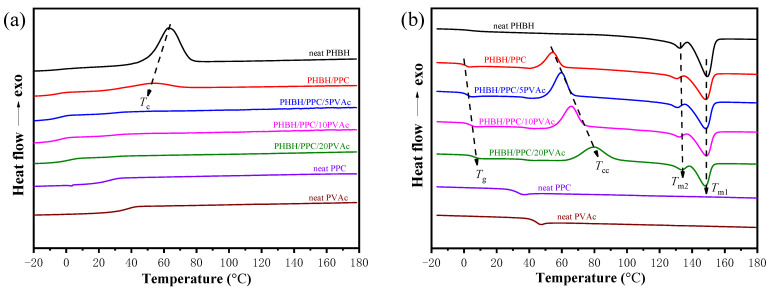
DSC thermograms of (**a**) first cooling at a cooling rate of 5 °C min^−1^ and (**b**) second heating at a heating rate of 10 °C min^−1^ for neat PHBH, PPC, PVAc, PHBH/PPC blends and PHBH/PPC/PVAc blends.

**Figure 7 polymers-15-04281-f007:**
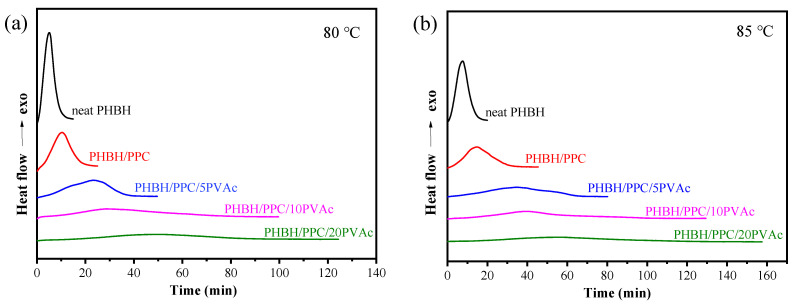
DSC curves for isothermal crystallization of neat PHBH and its blends at (**a**) 80 and (**b**) 85 °C.

**Figure 8 polymers-15-04281-f008:**
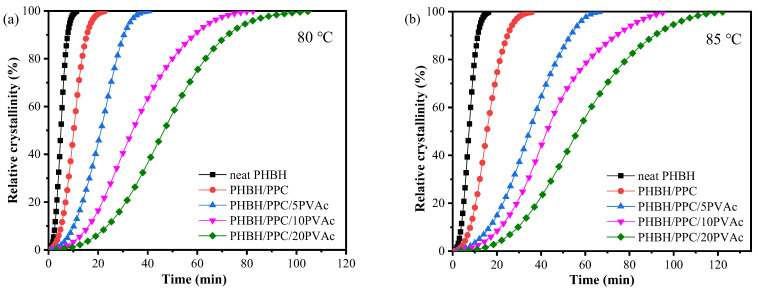
Relative crystallinity curves of neat PHBH and its blends at (**a**) 80 and (**b**) 85 °C.

**Figure 9 polymers-15-04281-f009:**
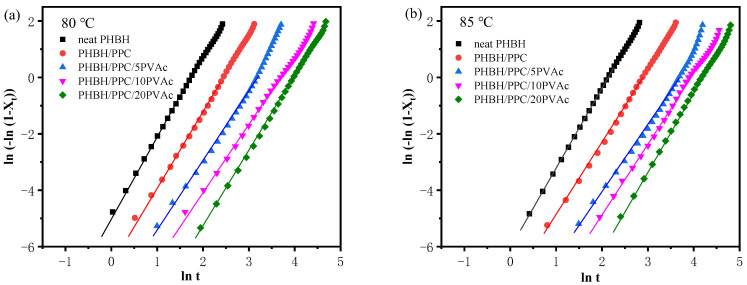
Avrami curves of neat PHBH and its blends at (**a**) 80 and (**b**) 85 °C.

**Figure 10 polymers-15-04281-f010:**
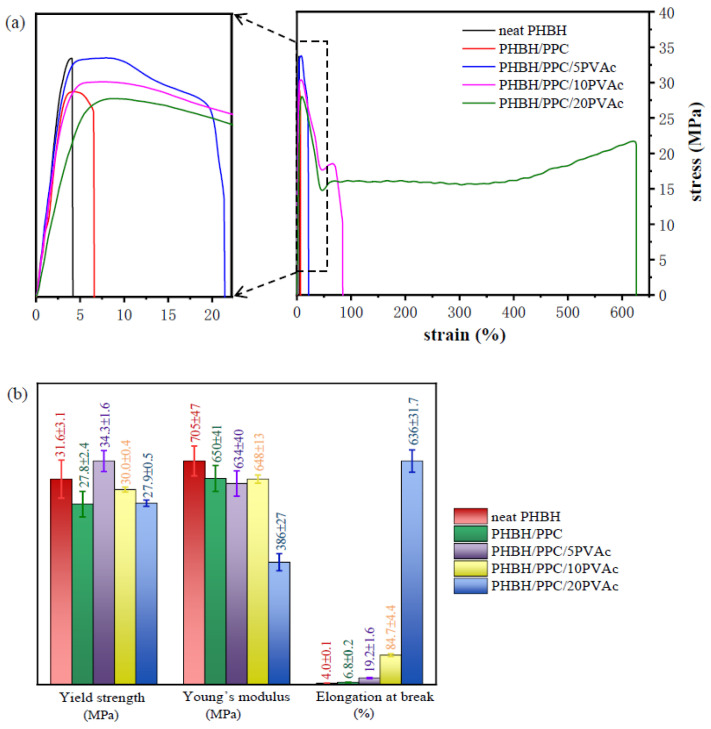
(**a**) Typical stress–strain curves of neat PHBH and its blends and (**b**) detailed tensile results about yield strength, Young’s modulus and elongation at break.

**Figure 11 polymers-15-04281-f011:**
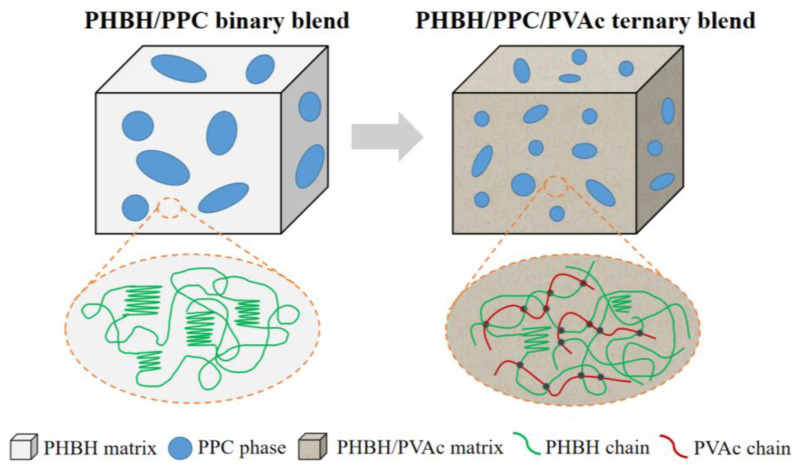
Schematic illustration of selective dispersion of PVAc in the ternary blends.

**Figure 12 polymers-15-04281-f012:**
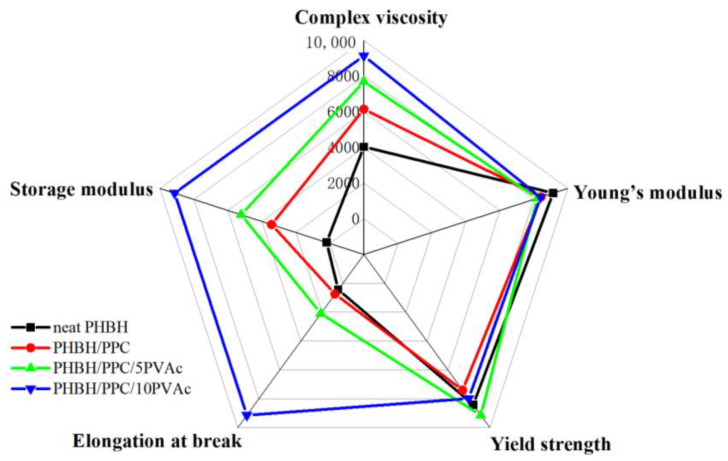
Comprehensive properties of neat PHBH, PHBH/PPC blends and PHBH/PPC/PVAc blends.

**Table 1 polymers-15-04281-t001:** Glass transition temperatures (*T_g_*s) of all samples from DMA tests.

Samples	*T*_*g*,PHBH_ (°C)	*T*_*g*,PPC_ (°C)	*T*_*g*,PVAc_ (°C)
neat PHBH	13.9	-	-
neat PPC	-	33.0	-
neat PVAc	-	-	47.9
PHBH/PPC	12.5	34.7	-
PHBH/PPC/5PVAc	16.8	35.2	-
PHBH/PPC/10PVAc	18.9	35.3	-
PHBH/PPC/20PVAc	Not detected	35.8	-
70PPC/30PVAc	-	38.9	-
50PPC/50PVAc	-	40.6	-
90PHBH/10PVAc	25.6	-	-
80PHBH/20PVAc	32.9	-	-
70PHBH/30PVAc	35.6	-	-

**Table 2 polymers-15-04281-t002:** Thermal properties of PHBH, PPC, PVAc, PHBH/PPC blends and PHBH/PPC/PVAc blends.

Sample	First Cooling		Second Heating						
	*T_c_* (°C)	Δ*H_c_* (J/g)	*T*_*g*,PHBH_ (°C)	*T_cc_* (°C)	Δ*H_cc_* (J/g)	*T*_*m*1_ (°C)	*T*_*m*2_ (°C)	Δ*H_m_* (J/g)	*X_c_* (%)
neat PHBH	62.8	49.7	2.8	-	-	132.5	149.3	55.9	38.3
neat PPC	-	-	33.3	-	-	-	-	-	
neat PVAc	-	-	44.7	-	-	-	-	-	
PHBH/PPC	53.0	17.6	1.6	54.4	24.7	130.5	148.4	49.0	16.6
PHBH/PPC/5PVAc	-	-	2.4	59.6	40.8	130.7	148.5	52.9	8.3
PHBH/PPC/10PVAc	-	-	4.3	65.8	43.7	132.8	148.9	49.4	3.9
PHBH/PPC/20PVAc	-	-	6.5	80.4	47.5	133.5	148.5	49.3	1.2

**Table 3 polymers-15-04281-t003:** Isothermal crystallization kinetic parameters of neat PHBH and its blends based on the Avrami equation.

Sample	Crystallization Temperature of 80 °C				Crystallization Temperature of 85 °C			
	*t*_1/2_ (min)	*n*	*k* (min^−n^)	*R* ^2^	*t*_1/2_ (min)	*n*	*k* (min^−n^)	*R* ^2^
PHBH	4.8	2.8	8.23 × 10^−3^	0.9998	7.2	2.8	2.36 × 10^−3^	0.9998
PHBH/PPC	10.2	2.7	1.39 × 10^−3^	0.9982	15.5	2.6	5.26 × 10^−4^	0.9986
PHBH/PPC/5PVAc	20.9	2.6	3.16 × 10^−4^	0.9940	35.3	2.4	1.20 × 10^−4^	0.9963
PHBH/PPC/10PVAc	34.1	2.4	1.63 × 10^−4^	0.9986	43.1	2.5	5.54 × 10^−5^	0.9982
PHBH/PPC/20PVAc	46.8	2.7	2.30 × 10^−5^	0.9994	56.9	2.8	8.81 × 10^−6^	0.9992

**Table 4 polymers-15-04281-t004:** Mechanical properties of neat PHBH and its blends.

Sample	Yield Strength (MPa)	Breaking Strength (MPa)	Young’s Modulus (MPa)	Elongation at Break (%)
neat PHBH	31.6 ± 3.1	31.6 ± 3.1	705 ± 47	4.0 ± 0.1
PHBH/PPC	27.8 ± 2.4	27.8 ± 2.4	650 ± 41	6.8 ± 0.2
PHBH/PPC/5PVAc	34.3 ± 1.6	34.3 ± 1.6	634 ± 40	19.2 ± 1.6
PHBH/PPC/10PVAc	30.0 ± 0.4	18.5 ± 1.2	648 ± 13	84.7 ± 4.4
PHBH/PPC/20PVAc	27.9 ± 0.5	20.8 ± 0.5	386 ± 27	636 ± 31.7

## Data Availability

The raw/processed data required to reproduce these findings are available upon request from the authors.

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
