# Peer review of "Ternary Blends from Biological Poly(3-hydroxybutyrate-*co*-3-hydroxyhexanoate), Poly(propylene carbonate) and Poly(vinyl acetate) with Balanced Properties"

_polymers, 2023, doi:10.3390/polym15214281_

Round 1

Reviewer 1 Report

Comments and Suggestions for Authors

The manuscript titled “Ternary Blends from Biological Poly(3-hydroxybutyrate-co-3-hydroxyhexanoate), Poly(propylene carbonate), and Poly(vinyl acetate) with Balanced Properties” presents an intriguing study on ternary polymer blends. The research topic is promising, and the results have the potential to contribute significantly to the field of polymer science. However, some critical issues need to be addressed before the manuscript can be considered for publication.

One of the fundamental issues in the manuscript is the lack of characterization of the chemical properties of the ternary blends using Fourier-Transform Infrared Spectroscopy (FTIR). FTIR is an essential analytical technique for elucidating the chemical interactions between polymer components and, subsequently, assessing the miscibility of the ternary blends. Without FTIR analysis, the authors miss an opportunity to provide crucial insights into the molecular-level interactions that may be occurring in their ternary blends.

The absence of FTIR data severely limits the scientific rigor of this study. FTIR analysis could have helped to identify functional groups, chemical bonding, and potential phase separation in the blends. Moreover, it would have provided critical evidence to support the claim of “better miscibility” in the manuscript, as chemical interactions often play a significant role in determining the properties of polymer blends.

I strongly recommend that the authors conduct FTIR analysis on their ternary blends and include the results in the manuscript. This analysis should be performed on individual polymer components and the ternary blends. The FTIR spectra can identify any new peaks, shifts in peaks, or changes in peak intensities, which would indicate interactions between the polymers. This is crucial for demonstrating the miscibility of the blends.

Once the FTIR analysis is performed, the authors should provide a comprehensive discussion of the results. They should interpret the spectra in terms of chemical interactions, such as hydrogen bonding or chemical reactions, that may be occurring in the blends. This discussion will help the readers understand the chemical basis for any observed property enhancements in the ternary blends.

If FTIR analysis reveals significant chemical interactions and miscibility in the ternary blends, the authors may want to consider revising the abstract and conclusion to reflect this important aspect of their study. A more accurate representation of the research could attract a wider readership and provide a better context for the results.

The authors should reevaluate their results and conclusions in light of the FTIR analysis. Any implications of chemical interactions on the mechanical, thermal, or other properties of the ternary blends should be discussed and incorporated into the manuscript.

Finally, the manuscript presents an interesting study on ternary polymer blends with the potential for balanced properties. The absence of FTIR analysis to characterize chemical interactions is a major gap that needs to be addressed before publication. The inclusion of FTIR data and a thorough discussion of its implications will significantly enhance the scientific quality and impact of this work.

Comments on the Quality of English Language

Considering the current manuscript, it is clear that moderate editing of English is crucial to improve readability, coherence, and clarity, addressing issues like grammar, sentence structure, word choice, and punctuation, all to communicate research findings to a wider audience effectively.

Author Response

Response

First of all, we really appreciate reviewer’s helpful comments. This is a very good comment. As the reviewer said, FTIR is an essential analytical technique for elucidating the chemical interactions between polymer components, and FTIR analysis could have helped to identify functional groups, chemical bonding, and potential phase separation in the blends. We have added FTIR tests and analyzed the spectra of all samples, as shown in Fig. S1.

As can be seen, the absorption peaks at 1720 cm-1 and 1044 cm-1 of neat PHBH corresponded to the stretching vibrations of C=O bond and C−O bond, respectively. It could be observed that there was no significant difference between PHBH/PPC/PVAc ternary blends and neat PHBH, which proved that there was no new reaction between PPC phase and PHBH matrix, and between PHBH and PVAc. The CH3···O=C interaction in the PHBH/PPC/PVAc ternary blends might be the molecular mechanisms to improve the miscibility of the blends, as reported in the literature (El-Hadi A. M. Polym. Eng. Sci. 2011, 51, 2191–2202). Unfortunately, the absorption peaks assigned to the deformation bands of CH3 and C=O were not observed in our work, probably due to the low PVAc concentration. The mechanism of miscibility in the PHBH/PPC/PVAc ternary blends will also be the focus of our next work.

As pointed out by the reviewer, FTIR analysis is crucial for demonstrating the miscibility of the blends due to that the miscibility between components is the basis of studying the balanced properties of PHBH/PPC/PVAc ternary blends. Therefore, we have added a discussion about the miscibility of PHBH/PVAc and PPC/PVAc binary blend systems by using DMA, as shown in Figure 1b and 1c on page 6. The results suggested that PVAc was miscible with both PHBH and PPC, and displayed better miscibility with PHBH compared with PPC.

Reviewer 2 Report

Comments and Suggestions for Authors

The paper illustrates improvement in mechanical and flow properties of PHBH/PPC/PVAc blends with respect to those of neat PHBH and of PHBH/PC blend. Because obtaining bio-based and biodegradable materials suitable for substitution of petroleum-based polymeric materials is of broad interest, the paper can be interesting for readers of Polymers. The paper contains some errors and not fully adequately discussed points which should be corrected.

r. 79 – instead of :...copolymers, and “reactive blending” ... should be: ...copolymers, or “reactive blending”...

Sec. 2.3.3 – it should be specified if PVAc is soluble in acetone

Fig. 2 – absolute value of complex viscosity of PHBH has maximum in dependence on the frequency. It is in discrepancy with classical behavior of viscoelastic melts. Determined dependence indicates degradation of PHBH or other problems during measurements. This point should be discussed.

r. 253-269 – an increase in viscosity of PHBH/PPC/PVAc blends with respect to PHBH/PPC blends is explained purely as a consequence of an increase of entanglement density in PHBH+PVAc matrix in comparison with neat PHBH. According to my feeling, also an increase in the entanglement density (decrease of the slip) at the interface between the matrix and PPC plays important role.

Fig. 3 – the caption is incorrect. Picture a) relates to neat PHBH and other designations should be shifted.

Fig. 4 – an increase in D for PHBH/PPC/5PVAc with respect to D for PHBH/PPC should be mentioned and briefly discussed.

Comments on the Quality of English Language

English should be checked and polished.

Author Response

Reviewer comment 2 and responses:

The paper illustrates improvement in mechanical and flow properties of PHBH/PPC/PVAc blends with respect to those of neat PHBH and of PHBH/PC blend. Because obtaining bio-based and biodegradable materials suitable for substitution of petroleum-based polymeric materials is of broad interest, the paper can be interesting for readers of Polymers. The paper contains some errors and not fully adequately discussed points which should be corrected.

Response

First of all, we really appreciate reviewer’s helpful comments. We have thoroughly revised the manuscript in accordance with the reviewer's suggestions.

  • 79 – instead of :...copolymers, and “reactive blending” ... should be: ...copolymers, or “reactive blending”...

Response

We appreciate the reviewer's comment. We have replaced “and ” with “or”. Please see page 2, line 83.

[Page 2, line 83] The most common approaches to improve the compatibility of immiscible polymer blends are the addition of block or graft copolymers, or “reactive blending.”

  • 2.3.3 – it should be specified if PVAc is soluble in acetone.

Response

We greatly appreciate the reviewer's valuable comment. The acetone solution is a good solvent for neat PVAc and neat PPC. However, neat PHBH is not soluble in acetone solution. In this study, PVAc was miscible with PHBH at the molecular level and PVAc was selectively dispersed in PHBH matrix. Therefore, the small amount of PVAc molecules dispersed in the PHBH matrix could not be solubilized in acetone solution. We have added a discussion of the solubility of PVAc in acetone solution in the manuscript. Please see page 12, line 319-323.

[Page 12, line 310-313] It was worth noting that acetone solution was a solvent for neat PVAc. From the previous discussion, PVAc was miscible with PHBH at the molecular level and was selectively localized in the PHBH matrix. Therefore, the small amount of PVAc dispersed in the PHBH matrix could not be solubilized in acetone solution.

  • absolute value of complex viscosity of PHBH has maximum in dependence on the frequency. It is in discrepancy with classical behavior of viscoelastic melts. Determined dependence indicates degradation of PHBH or other problems during measurements. This point should be discussed.

Response

Thank you for the comment. Neat PHBH in the present work showed a Newtonian plateau at low frequency region and had the smallest complex viscosity of the three neat components. From Fig. 3c, the decreased complex viscosity of neat PHBH at the low-frequency region may be due to the degradation of PHBH during testing at 155 ℃. We have added the discussion on the dependence of viscosity on frequency in the manuscript, as reviewer suggested. Again, thank you for the reviewer’s valuable comment.

[Page 9, line 256-260] As can be seen from Figure 3c, neat PHBH had a Newtonian plateau at intermediate frequency region with the smallest viscosity of the three neat components. In the low-frequency region, the decreasing complex viscosity with decreasing frequency might be due to the slight degradation of PHBH during testing at 155 ℃.

  • 253-269 – an increase in viscosity of PHBH/PPC/PVAc blends with respect to PHBH/PPC blends is explained purely as a consequence of an increase of entanglement density in PHBH+PVAc matrix in comparison with neat PHBH. According to my feeling, also an increase in the entanglement density (decrease of the slip) at the interface between the matrix and PPC plays important role.

Response

We greatly appreciate the reviewer's valuable comment. We strongly agreed with the reviewer's comments and have revised this section, as suggested by the reviewer.

[page 10, line 282-288] adding PVAc into PHBH/PPC blend resulted in an increase in the entanglement density in the continuous phase and at the interface between the PPC dispersion phase and the PHBH/PVAc continuous phase. The enhancement of entanglement density of PHBH matrix, and between matrix and dispersion phase resulted in an increase in melt viscosity and elasticity.

  • 3 – the caption is incorrect. Picture a) relates to neat PHBH and other designations should be shifted.

Response

We thank for the reviewer’s comment. We have modified the caption of Fig. 4.

[Page 11, Figure 4] SEM images of surfaces for: (a) neat PHBH, (b) PHBH/PPC, (c) PHBH/PPC/5PVAc, (d) PHBH/PPC/10PVAc, and (e) PHBH/PPC/20PVAc.

  • 4 – an increase in D for PHBH/PPC/5PVAc with respect to D for PHBH/PPC should be mentioned and briefly discussed.

Response

We thank for the reviewer’s comment. As noted by the reviewer, D value of PHBH/PPC/5PVAc deviated a little from that observed in the SEM shown in Fig. 4. We apologize for the reversal of Fig. 4a and Fig. 4b. We again tested the average particle size of the dispersed phase of the blends via analysis of Nano Measurer 1.2 software. An average of 50-100 particles from SEM images was measured for each sample. The calculated dispersed phase sizes were (1.68 ± 0.43) μm for PHBH/PPC, (1.38 ± 0.34) μm for PHBH/PPC/5PVAc, (1.03 ± 0.31) μm for PHBH/PPC/10PVAc, and (0.84 ± 0.22 ) μm for PHBH/PPC/20PVAc. The dispersed phase size of blends was decreased with the increasing PVAc content. We have modified the expression of the size of the dispersed phase of the blends.

  • [Page 12, Figure 5] The average size of dispersed phase (D) of PHBH/PPC and PHBH/PPC/PVAc blends.
  • English should be checked and polished

Response

We really appreciate reviewer’s helpful comments. We have thoroughly revised and polished English of this manuscript in accordance with the reviewer's suggestions.

Reviewer 3 Report

Comments and Suggestions for Authors

In this paper, blending with PPC is investigated to solve the moldability and brittleness of PHBH, and PVAc is reported as an effective compatibilizer. The experimental results and discussion are consistent and agree with the compatibilization mechanism discussed. I believe that the following minor modifications will improve the quality of the paper.

Scheme 1: (a) PHBH, (b) PPC, and (c) PVAc.

Table 1: How about "Not detected" instead of "-" for Tg,PHBH in Table1 PHBH/PPC/20PVAc?

Figure 3: The explanation in the caption does not correspond to the numbers in the figure. Is (a) a picture of PHBH/PPC before etching?

Figure 4: Please add the standard deviation in the figure.

Figure 5: Please indicate in the figure what peaks and base shifts correspond to what. 

Figure 7 and 8 are different from the other figures in terms of color. Please unify them.

PVAc is a highly water-absorbent polymer, and its Tg changes depending on the moisture content. Did you adjust the moisture content before conducting the tensile test? Please add this information to the experimental method.

Table 4: When the PVAc content increases from 10 to 20 wt%, the elastic modulus drops all at once and the elongation at break increases significantly. Please discuss this mechanism additionally. I believe that factors such as the particle size distribution and the effect of water content in PVAc are possible factors.

I am not sure if PHBH/PVAc would also solve the problem, although it is not discussed in this paper. It seems to be a highly compatible combination, so I was wondering if it would be possible to make more stable adjustments.

Author Response

Reviewer comment 3 and responses:

In this paper, blending with PPC is investigated to solve the moldability and brittleness of PHBH, and PVAc is reported as an effective compatibilizer. The experimental results and discussion are consistent and agree with the compatibilization mechanism discussed. I believe that the following minor modifications will improve the quality of the paper.

Response

First of all, we really appreciate reviewer’s helpful comments. We have thoroughly revised the manuscript in accordance with the reviewer's suggestions.

Scheme 1: (a) PHBH, (b) PPC, and (c) PVAc.

Response

We appreciate the reviewer's comment. We have revised the caption of Scheme 1 in accordance with the reviewer's suggestion.

[Page 3, Scheme 1] Scheme 1. Chemical structures of (a) PHBH, (b) PPC, and (c) PVAc.

  • Table 1: How about "Not detected" instead of "-" for Tg,PHBH in Table1 PHBH/PPC/20PVAc? 

Response

We thank for the reviewer’s comment. We have replaced “-” with “Not detected” in Table 1 for Tg of PHBH in PHBH/PPC/20PVAc. Please see page 8, Table1.

  • Figure 3: The explanation in the caption does not correspond to the numbers in the figure. Is (a) a picture of PHBH/PPC before etching?

Response

We apologize for the error in the caption of Figure 4. We have modified the caption of Figure 4.

[Page 11, Figure 4] SEM images of surfaces for: (a) neat PHBH, (b) PHBH/PPC, (c) PHBH/PPC/5PVAc, (d) PHBH/PPC/10PVAc, and (e) PHBH/PPC/20PVAc.

  • Figure 4: Please add the standard deviation in the figure. 

Response

We thank for the reviewer’s comment. We have modified the expression of the size of the dispersed phase of the blends and added the standard deviation in Figure 5. Please see Figure 5 on page 12.

  • Figure 5: Please indicate in the figure what peaks and base shifts correspond to what.  

Response

We greatly appreciate the reviewer's valuable comment. In response, we have included arrows indicating the direction of peak shift in Figure 6 in the revised manuscript. Please see Figure 6 on page 13.

  • Figure 7 and 8 are different from the other figures in terms of color. Please unify them.

Response

Thanks for the reviewer’s comment. We have revised color of Figure 8 and Figure 9  in accordance with the reviewer's suggestion. Please see Figure 8 and 9 on page 16.

  • PVAc is a highly water-absorbent polymer, and its Tg changes depending on the moisture content. Did you adjust the moisture content before conducting the tensile test? Please add this information to the experimental method. 

Response

We appreciate the reviewer’s comment. PVAc has ester groups on the side chain and is free of hydroxyl and carboxyl groups. Its molecular structure is similar to PLA, as shown in Figure 1. The water absorption of PVAc is 2%-5%. The water absorption of PVAc is slightly higher compared to traditional petroleum-based plastics, but similar to other biodegradable ones. For example, PP, PS, PC and PET have water absorption of less than 1%. However, PLA and PPC have water absorption of 2.8% and 4.4%, respectively. The water absorption of PVAc is similar to that of PPC. Therefore, in the present work, we did not consider the effect of water absorption of PVAc on the performance of the blends. During the preparation of the blends, PHBH, PPC, and PVAC were dried in a vacuum drying oven at 80 °C for 10 h before processing, in order to avoid water absorption. We did not dry the samples prior to tensile and thermal property testing.

  • Table 4: When the PVAc content increases from 10 to 20 wt%, the elastic modulus drops all at once and the elongation at break increases significantly. Please discuss this mechanism additionally. I believe that factors such as the particle size distribution and the effect of water content in PVAc are possible factors. 

Response

Thank you for the reviewer's comment. For the ternary blends with 20 wt% PVAc, the decrease in elastic modulus might be due to the decrease in the degree of crystallinity. Due to the dilution effect of amorphous PVAc, the degree of crystallinity of the blends decreased with the increase of PVAc content, which decreased to 1.2% at 20 wt% PVAc, resulting in a decrease in modulus and an increase in elongation at break. The effect of water absorption of PVAc on the mechanical properties, crystallization and thermal behavior, and degradation rates of the blends is also a very interesting scientific issue. We will next design experiments to discuss this issue. Thanks again for your valuable suggestions and comments!

(8) I am not sure if PHBH/PVAc would also solve the problem, although it is not discussed in this paper. It seems to be a highly compatible combination, so I was wondering if it would be possible to make more stable adjustments.

Response

We greatly appreciate the reviewer's valuable comment. The acetone solution is a good solvent for neat PVAc and neat PPC. However, neat PHBH is not soluble in acetone solution. In this study, PVAc was miscible with PHBH at the molecular level and PVAc was selectively dispersed in PHBH matrix. Of course, if a large amount of PVAc was introduced, resulting in the migration of PVAc phase from the PHBH matrix into the PPC dispersed phase or to the interface, it can also be etched away by the acetone solution. Therefore, the small amount of PVAc molecules dispersed in the PHBH matrix could not be solubilized in acetone solution. We have added a discussion of the solubility of PVAc and PHBH/PVAc phase in acetone solution in the manuscript. Please see page 11-12, line 314-320.

[Page 11 and 12, line 307-313] For the PHBH/PPC and PHBH/PPC/PVAc blends (Figure 4b-4e), the dispersed PPC phase etched away by acetone left homogeneous and spherical voids, implying that PHBH matrix and PPC dispersed phase were immiscible, which was in line with the observation of DMA. It was worth noting that acetone solution was a solvent for neat PVAc. From the previous discussion, PVAc was miscible with PHBH at the molecular level and was selectively localized in the PHBH matrix. Therefore, the small amount of PVAc dispersed in the PHBH matrix could not be solubilized in acetone solution.

Reviewer 4 Report

Comments and Suggestions for Authors

Authors reported a very interesting result entitled “Ternary Blends from Biological Poly(3-hydroxybutyrate-co-3-2 hydroxyhexanoate), Poly(propylene carbonate) and Poly(vinyl 3 acetate) with Balanced Properties ”. I think the result is publishable if the below issue is cleared.

-        In the case of binary systems, Authors focused on PHBH/PPC only. Would you report the data (e.g., Tg) for the binary systems such as PHBH/PVAc and PPC/PVAc for clarity?

-        Authors mentioned that “the PVAc phase in the PHBH/PPC/PVAc ternary blends was full miscible with PHBH matrix and was selectively localized in the PHBH matrix, acting as compatibilizer."  For clarifying it, PHBH/PVAc binary system’s data should be needed.

Comment: Considering the critical chi-parameter value for the binary polymer-polymer system, the chi-parameter should be smaller than the critical chi-parameter for polymer-polymer miscibility.

In general, two polymers are usually immiscible if there is no specific interaction (and/or polarity should be very similar each other), indicating the chi-parameter is very small. Do you have the surface polarity data for PHBH and PVAc for checking each polarity of polymers?

-        Additional minor comment:

(1)   The unit for molecular weight: “~ kg mol-1” may be better than “g mol-1” because GPC is not much sensitive in general.

(2)   (Lines 106-109) "Polydispersity index" may be abbreviated as PDI as usual.

Comments on the Quality of English Language

A minor revision is required.

Author Response

Reviewer comment 4 and responses:

Authors reported a very interesting result entitled “Ternary Blends from Biological Poly(3-hydroxybutyrate-co-3-2 hydroxyhexanoate), Poly(propylene carbonate) and Poly(vinyl 3 acetate) with Balanced Properties ”. I think the result is publishable if the below issue is cleared.

Response

First of all, we really appreciate reviewer’s helpful comments. We have thoroughly revised the manuscript in accordance with the reviewer's suggestions.

  • In the case of binary systems, Authors focused on PHBH/PPC only. Would you report the data (e.g., Tg) for the binary systems such as PHBH/PVAc and PPC/PVAc for clarity?

Response

Thanks for the reviewer’s comment. We have included the Tg data of PPC/PVAc binary and PHBH/PPC binary blend system, as shown in Figure 1b and 1c. As can be seen from damping factor curves of PPC/PVAc binary and PHBH/PPC binary blends, only one tan δ peak between those of neat components was observed and it increased with increasing PVAc content. These results suggested that PVAc was miscible with both PHBH and PPC.

[Page 7, line 214-221] As can be seen from Figure 1b, PPC/PVAc binary blends showed only one tan δ peak, which was between those of neat PPC and PVAc and increased with PVAc content. Variation of Tg of PHBH/PVAc binary blends was similar to that of PPC/PVAc blends, as shown in Figure 1c. These results demonstrated that PVAc was miscible with both PHBH and PPC, and displayed better miscibility with PHBH compared with PPC. Therefore, it can be inferred that PVAc was probable selectively localized in the PHBH matrix, increasing the Tg of PHBH in PHBH/PPC/PVAc blends.

  • Authors mentioned that “the PVAc phase in the PHBH/PPC/PVAc ternary blends was full miscible with PHBH matrix and was selectively localized in the PHBH matrix, acting as compatibilizer." For clarifying it, PHBH/PVAc binary system’s data should be needed.

Response 

Thanks for the comment. We included the Tg data of PHBH/PVAc binary blends by using DMA tests. There was only one Tg between those of pure PHBH and PPC indicating that PVAc phase was full miscible with PHBH matrix.

[Page 7, line 216-220] Variation of Tg of PHBH/PVAc binary blends was similar to that of PPC/PVAc blends, as shown in Figure 1c. These results demonstrated that PVAc was miscible with both PHBH and PPC.

  • Comment: Considering the critical chi-parameter value for the binary polymer-polymer system, the chi-parameter should be smaller than the critical chi-parameter for polymer-polymer miscibility.

In general, two polymers are usually immiscible if there is no specific interaction (and/or polarity should be very similar each other), indicating the chi-parameter is very small. Do you have the surface polarity data for PHBH and PVAc for checking each polarity of polymers?

Response

We greatly appreciate the reviewer's valuable comment. We strongly agree with the reviewer that the surface polarities of polymers can determine their miscibility. To obtain the surface polarity of PHBH and PVAc, the static contact angles (CA) of water on PHBH and PVAc films were measured on a DSA 100S static contact angle measurement instrument at ambient temperature. The mean value of the CAs recorded on each sample was taken as the final result. The change of static water contact angle is displayed in Figure 1S. The film made by raw PHBH granules has a hydrophilic surface with a CA of 78°. The CA on the film made by PVAc granules is 73°, indicating similar surface polarity between PHBH and PVAc.

Additional minor comment:

  • The unit for molecular weight: “~ kg mol-1” may be better than “g mol-1” because GPC is not much sensitive in general.

Response

Thank you for the comment. We have changed the unit of molecular weight to “kg mol-1”, as suggested by the reviewer.

[Page 3, line 110-113] It had a weight-average molecular weight (Mw) and polydispersity (PDI) of 754 kg mol−1 and 4.89, respectively. PHBH copolymer had 6 mol% 3HH unit based on 1H NMR. PPC was provided by Changchun Institute of Applied Chemistry. It had an Mw of 130 kg mol−1 and PDI of 2.6. PVAc with Mw of 315 kg mol−1 and PDI of 1.9 was bought from Nuoda New Materials Company (Yantai, China).

  • (Lines 106-109) "Polydispersity index" may be abbreviated as PDI as usual.

Response

We thank for the reviewer’s comment. We have abbreviated "Polydispersity index" as “PDI”.

[Page 3, line 110-114] It had a weight-average molecular weight (Mw) and polydispersity (PDI) of 754 kg mol−1 and 4.89, respectively. PHBH copolymer had 6 mol% 3HH unit based on 1H NMR. PPC was provided by Changchun Institute of Applied Chemistry. It had an Mw of 130 kg mol−1 and PDI of 2.6. PVAc with Mw of 315 kg mol−1 and PDI of 1.9 was bought from Nuoda New Materials Company (Yantai, China).

Round 2

Reviewer 1 Report

Comments and Suggestions for Authors

I am pleased to note that, in response to the previous review, the authors have now incorporated FTIR analysis for characterizing the chemical properties of the ternary blends. This is a significant and commendable addition to the manuscript. The inclusion of FTIR analysis strengthens the scientific foundation of the study, providing valuable insights into the chemical interactions within the blends, which is essential for understanding the miscibility and properties of these ternary blends. This improvement showcases the authors' dedication to enhancing the quality and impact of their research.

Comments on the Quality of English Language

The quality of the English language in this manuscript is generally good. However, there are a few areas that require minor editing to improve clarity and readability.